# Tenofovir, Another Inexpensive, Well-Known and Widely Available Old Drug Repurposed for SARS-COV-2 Infection

**DOI:** 10.3390/ph14050454

**Published:** 2021-05-11

**Authors:** Isabella Zanella, Daniela Zizioli, Francesco Castelli, Eugenia Quiros-Roldan

**Affiliations:** 1Department of Molecular and Translational Medicine, University of Brescia, 25123 Brescia, Italy; daniela.zizioli@unibs.it; 2Clinical Chemistry Laboratory, Cytogenetics and Molecular Genetics Section, Diagnostic Department, ASST Spedali Civili di Brescia, Piazzale Spedali Civili 1, 25123 Brescia, Italy; 3University Department of Infectious and Tropical Diseases, University of Brescia and ASST Spedali Civili di Brescia, Piazzale Spedali Civili 1, 25123 Brescia, Italy; francesco.castelli@unibs.it (F.C.); eugeniaquiros@yahoo.it (E.Q.-R.)

**Keywords:** tenofovir, nucleoside/nucleotide analogues, RNA-dependent RNA polymerase inhibitors, nucleotide SARS-CoV-2, COVID-19 treatment

## Abstract

Severe acute respiratory syndrome coronavirus 2 (SARS-CoV-2) infection is spreading worldwide with different clinical manifestations. Age and comorbidities may explain severity in critical cases and people living with human immunodeficiency virus (HIV) might be at particularly high risk for severe progression. Nonetheless, current data, although sometimes contradictory, do not confirm higher morbidity, risk of more severe COVID-19 or higher mortality in HIV-infected people with complete access to antiretroviral therapy (ART). A possible protective role of ART has been hypothesized to explain these observations. Anti-viral drugs used to treat HIV infection have been repurposed for COVID-19 treatment; this is also based on previous studies on severe acute respiratory syndrome virus (SARS-CoV) and Middle East respiratory syndrome virus (MERS-CoV). Among them, lopinavir/ritonavir, an inhibitor of viral protease, was extensively used early in the pandemic but it was soon abandoned due to lack of effectiveness in clinical trials. However, remdesivir, a nucleotide analog that acts as reverse-transcriptase inhibitor, which was tested early during the pandemic because of its wide range of antiviral activity against several RNA viruses and its safety profile, is currently the only antiviral medication approved for COVID-19. Tenofovir, another nucleotide analog used extensively for HIV treatment and pre-exposure prophylaxis (PrEP), has also been hypothesized as effective in COVID-19. No data on tenofovir’s efficacy in coronavirus infections other than COVID-19 are currently available, although information relating to SARS-CoV-2 infection is starting to come out. Here, we review the currently available evidence on tenofovir’s efficacy against SARS-CoV-2.

## 1. Introduction

COVID-19 (coronavirus disease 2019), caused by the severe acute respiratory syndrome virus 2 (SARS-CoV-2), has been defined as a pandemic around the world by the World Health Organization (WHO). Millions of people are currently infected and the disease causes death in 5–10% of patients. COVID-19 was first reported in December 2019, and at that time we faced a disease without available drugs for treatment or prevention. Antiviral therapies are a key medical countermeasure for emerging viral infections because vaccine production takes a long time whereas effective antiviral medications may already exist. Further, even though several vaccines have been proven effective in providing protection against COVID-19 or at least in preventing serious illness and death and they have been fully approved for use in humans by the appropriate authorities in many parts of the world [1,2,3], it is unknown whether they can provide long-term protection, are completely or only partly efficacious against the new emerging virus variants and will soon be available in low-resource settings around the world. Therefore, also considering that some patients cannot be vaccinated because of health conditions, there is still an urgent need for efficacious and easily accessible therapies against the pandemic disease. For COVID-19, the potential of several existing medications (including remdesivir, lopinavir/ritonavir, hydroxychloroquine and tocilizumab) have been of interest based on previous studies of well-known coronavirus family members: severe acute respiratory syndrome virus (SARS-CoV) and Middle East respiratory syndrome virus (MERS-CoV). One approach to treat viral infection is targeting viral functional proteins [4]. Like SARS-CoV and MERS-CoV, SARS-CoV-2 is an enveloped positive-sense single-stranded RNA virus. Viral particles consist of structural proteins that surround and interact with the viral genome, namely, the spike (S), envelope (E), membrane (M) and nucleocapsid (N) proteins. The virus infects target cells after specific binding of the receptor-binding domain (RBD) of the viral protein S to the cellular entry receptor, the angiotensin-converting enzyme 2 (ACE2). Following entry, virions are uncoated, viral RNA is released and immediately translated into two large polyproteins (pp1a and pp1a), specified by two large open reading frames of the viral RNA (ORF1a and ORF1b). These polyproteins are then cleaved by two viral proteases, the papain-like protease (PLpro) and the main protease or 3C-like protease (Mpro/3CLpro), releasing 16 non-structural proteins (nsp1–16). Most of these proteins are involved in the intracellular replication cycle of SARS-CoV-2, in particular, nsp12 is the RNA dependent RNA polymerase (RdRp) that with the cofactors nsp7 and nsp8 performs viral RNA synthesis for translation and synthesis of structural and accessory proteins and for the assembly of new virions [4,5,6] (Figure 1). 

SARS-CoV-2 shares 82% RNA sequence identity with SARS-CoV and MERS-CoV, therefore, at the beginning of the pandemic, known drugs targeting viral RdRp of SARS-CoV or other coronaviruses were repurposed due to the urgent need for an antiviral therapy for the new member of the family, SARS-CoV-2. Among them, remdesivir (an adenosine analog) was tested early during the pandemic because of its wide range of antiviral activity against several RNA viruses (including Ebola virus, SARS-CoV and MERS-CoV) and its safety profile. Currently, it is the only antiviral medication approved for COVID-19 [7]. Early during the COVID-19 pandemic, lopinavir/ritonavir, an old drug used for human immunodeficiency virus (HIV) treatment, was also extensively used as a potential therapy against the new virus [8]. Lopinavir is a type 1 aspartate protease inhibitor with in vitro and in vivo activity against SARS-CoV and in an animal model against MERS-CoV [9]; however, after the first wave of COVID-19 it was soon abandoned due to its lack of effectiveness in clinical trials [10,11,12].

Generally, immunocompromised patients are more susceptible to bacterial, fungal, viral, and parasitic infections than healthy persons due to their inability to mount successful immune responses. For these reasons, at the beginning of pandemic, HIV-infected patients, who constitute 0.5% of the world population [13] were considered as high risk for severe COVID-19 [14,15], and if their antiviral therapy could be shifted to lopinavir without decreasing efficacy against HIV infection, it was done during the first wave [8,15,16,17]. Today, lopinavir is no longer used for COVID-19 treatment outside of clinical trials [18].

Up until now, it is not clear whether HIV infection increases COVID-19 risk in settings with complete access to antiretroviral drugs (ARVs) against HIV [19,20,21,22,23]. The concern over increased risk for severe COVID-19 disease for HIV-infected patients may be based on the assumption that these subjects are more likely to be immunosuppressed because HIV infection is associated with abnormal humoral and T-cell–mediated immune responses, resulting in increased susceptibility to numerous opportunistic infections. There may also be a higher prevalence of co-morbidities that appear to be driving factors for COVID-19 mortality such as hypertension, diabetes, chronic lung disease, serious cardiovascular conditions, chronic kidney disease, or chronic liver disease or cancer [24,25].

It is not known if people living with HIV (PLWH), who are clinically and virologically stable, experience any greater risk for COVID-19 complications than the population without HIV infection [19,20,21,22,23,26,27,28,29,30,31,32,33,34,35,36,37,38,39,40,41,42], and there is a possible protective role of antiretroviral therapy (ART) [43,44]. At present, the available data mainly appear in case reports and case series of SARS-CoV2/HIV co-infected patients [44,45,46,47]. 

The nucleotide analogue tenofovir, which was originally designed to inhibit the ATP polymerization into the growing nucleic acids chain by the HIV reverse transcriptase [48,49], has also been hypothesized to be effective in COVID-19 [50]. Tenofovir is commercially available as two prodrugs, tenofovir disoproxil fumarate (TDF), synthesized to increase tenofovir oral bioavailability, and tenofovir alafenamide (TAF), later designed to decrease the risk of TDF renal toxicity and bone density changes and to increase intestinal and plasma stability and intracellular accumulation, thereby reducing dosage [51] (Figure 2).

After oral administration, TDF is hydrolyzed by gut and plasma esterases to tenofovir, and TAF is metabolized mostly intracellularly by cathepsin A to tenofovir. Tenofovir is then intracellularly activated to tenofovir-diphosphate, its triphosphate active form [52]. Acting as adenosine nucleotide analogues, TDF and TAF mimic the structure of the natural nucleotide ATP, and are recognized by the HIV reverse transcriptase or the hepatitis B virus (HBV) polymerase/reverse transcriptase as substrate, acting as chain terminators and then blocking viral replication.

TFD and TAF are now among the most popular antiviral drugs for the treatment of both HIV and HBV infections. For HIV infection, these drugs are recommended by several guidelines in combination antiretroviral therapy (cART) together with one or two further drugs, such as emtricitabine (FTC) and/or lamivudine (3TC) [53,54,55,56]. TDF and TAF are also extensively and efficaciously used as pre-exposure prophylaxis (PrEP) against HIV infection in people at risk of acquiring this infection [57,58,59,60,61]. The current first line therapies for chronic HBV infection also includes TDF and TAF, which have proven to be well-tolerated and also highly effective in preventing liver fibrosis progression [62,63,64]. Further, tenofovir has been proposed to reduce the risk of herpes simplex virus 1 and 2 (HSV-1, HSV-2) infections, although studies have shown conflicting results [65,66,67,68].

As stated above, TDF and TAF are also used to prevent HIV infection and several studies have demonstrated that selection of drug resistance (mostly due to undiagnosed acute HIV infection during PrEP, low adherence to prophylaxis or rare HIV acquisition despite PrEP) is less frequent for these molecules in comparison with other antiretroviral drugs like FTC [69]. Further, no TDF resistant strains have been observed after eight years of therapy in patients with chronic hepatitis B [70], although a genotypic resistance to tenofovir due to a quadruple mutation has been recently described in two chronic hepatitis B patients [71]. To date, we found no reports in the literature regarding development of tenofovir resistance to SARS-CoV-2 infection.

It has been discussed whether the nucleotide analogue tenofovir, which has a similar structure to remdesivir, may also achieve SARS-Cov-2 inhibition [72]. Here, we review the currently available evidence on tenofovir efficacy against SARS-CoV-2.

## 2. Methodology and Literature Search Strategy

### 2.1. Literature Search

A comprehensive search of PubMed, EMBASE and Cochrane Library was performed up to 12 April 2021 and was restricted to English, Italian and Spanish. Unpublished trials were also identified from the clinical trial registry platforms (http://clinicaltrials.gov/ accessed on 12 April 2021). The methodology of search followed the Preferred Reporting Items for Systematic Reviews and Meta-Analysis (PRISMA) guidelines (Figure 3). A manual search was conducted by screening the reference lists of inclusive studies. Our search strategy included the following relevant terms: “tenofovir AND COVID-19”, “tenofovir AND COVID19”, “TDF AND COVID-19”, “TDF AND COVID19”, “TAF AND COVID-19”, “TAF AND COVID19”, “tenofovir AND coronavirus”, “RNA polymerase RNA dependent inhibitors AND COVID-19”, “RNA polymerase RNA dependent inhibitors AND COVID19, “RNA polymerase RNA dependent inhibitors AND coronavirus”, “RNA polymerase RNA dependent inhibitors AND COVID-19 AND pre-clinic”, “animal model AND SARS-CoV-2 AND tenofovir”, “SARS-CoV-2 AND tenofovir”, “SARS-CoV-2 AND TDF”, “SARS-CoV-2 AND TAF”.

### 2.2. Selection of Studies

Two authors (E.Q.-R. and I.Z.) independently screened the titles, abstracts and full texts of retrieved articles to evaluate their eligibility (Figure 3). Reviews, case reports and non-peer-reviewed preprint articles were not considered in this review. In vitro and in vivo pre-clinical studies, randomized controlled trials, prospective and retrospective cohort studies, case series and clinical cases performed among HIV-infected adults with COVID-19 (only studies with more than 100 HIV-infected patients) were included in the current literature review. We reviewed studies from the literature about RdRp inhibitors use in COVID-19, mainly focusing on in vitro and in vivo efficacy, potential in silico efficacy, clinical outcome, mortality rate, virological eradication, safety and tolerability.

## 3. Pre-Clinical Studies

### 3.1. In Silico Studies

In silico-based screening of drugs that were approved for other indications and with proven safety profiles or of new molecules is a thoroughly applied and cost-effective strategy in the identification of potential therapies for COVID-19. The targets of this approach are the main proteins that SARS-CoV-2 uses to enter the cells, the spike protein S and its ACE2 receptor, or those needed for fusion, replication and spread, like RdRp, Mpro/3CLpro or PLpro (see Figure 1).

Some in silico studies reported that tenofovir may act as an inhibitor of some of these proteins, suggesting its use in COVID-19 therapy. Since tenofovir is a nucleotide analogue, most of these studies are focused on its potential inhibitory activity on SARS-CoV-2 RdRp. The first molecular docking (MD) study considering tenofovir in its triphosphate active form as a potential drug against SARS-CoV-2 RdRp revealed a good docking score value, related to 5 hydrogen bonds of the nucleotide analogue with crucial residues of the RdRp catalytic site that was reflected in a high binding energy value, suggesting the molecule as a promising therapeutic approach in COVID-19 [50]. Similar findings were obtained by Udofia et al. [73], using Sars-CoV-2 and Sars-CoV RdRp and Mpro/3CLpro as targets for 173 compounds and performing molecular docking in which the protein structures were kept rigid while ligands were allowed full rotations. Poustforoosh and colleagues [74] virtually screened 28 compounds against two active sites of RdRp by rigid and induced-fit docking, and found that tenofovir had lower docking scores than other tested drugs. Similar findings were obtained by Tiwari [75]. In a further study by Elficky [76], the researcher applied a molecular dynamics simulation (MDS) approach to test the binding affinity of several compounds to different dynamic states of the RdRp protein. Using a flexible ligand-flexible bonding site approach, MD of the selected compounds with the different conformations of RdRp used as target confirmed a good binding energy value for tenofovir. This value was not so different from those of the four natural nucleotide triphosphates (NTPs), suggesting that this drug may efficiently compete with natural NTPs for the binding site of SARS-CoV-2 RdRp, then possibly inhibiting the enzyme. To increase the degree of reliability of MD, Grahl and colleagues [77] used three different docking software to virtually screen 48 antiviral drugs against different SARS-CoV-2 proteins (S protein in different conformation states, its isolated receptor binding domain, Mpro/3CLpro, RdRp, nsp10 and nsp16). Combining data derived from the three analyses by considering both lipophilic and hydrophilic matches, the authors selected 18 protein-ligand complexes for further stability and binding strength analysis. Among these potential ligands, they found tenofovir as interacting with RdRp, although further MDS analyses were conducted only with the first three ranked drugs, penciclovir, ribavirin and zanamivir. Copertino et al. [78] recently described a comprehensive in silico analysis of the binding of several Food and Drug Administration (FDA)-approved antiretroviral drugs to the substrate binding site of the Mpro/3CLpro and RdRp of SARS-CoV-2 at atomic resolution. The authors applied an induced fit docking algorithm to calculate docking scores of drugs and MDS with a molecular mechanics-generalized Born surface area (MM-GBSA) approach to calculate the binding free energy and assess the stability of drug binding to the catalytic sites of both proteins. Interestingly, they found that tenofovir in its active triphosphate form, although it had docking score lower than other nucleoside/nucleotide analog reverse-transcriptase inhibitors (NRTIs) like abacavir (ABC), FTC and zidovudine (ZDV), showed more stable binding over time with the RdRp catalytic site, and was the only one to directly interact and bind with the residue D760 within the nucleotide binding site of the enzyme, suggesting that this NRTI could be an important drug candidate against SARS-Cov-2 infection. Using a similar approach, Hasan et al. [79] screened a library of 2400 compounds comprising approved RdRp inhibitor drugs, including tenofovir, and their structural analogues, in order to identify new potential inhibitors of the viral enzyme. The authors performed an initial virtual screening and validated the results by MD, using a semi-flexible docking approach, to identify the best compounds. By MDS and MM-GBSA analyses, the authors found that most of the interactions of those ligands with the enzyme were hydrophobic and involved residues R523, A524, R525 and R594, and that a structural analogue of tenofovir had the highest binding affinity. The researchers concluded also that based on computer aided absorption, distribution, metabolism, elimination and toxicity (ADMET) profiling analysis, these analogues may have therapeutic potential against SARS-CoV-2. To investigate potential combination ART, Dallocchio et al. [80] performed molecular docking of four HIV protease inhibitors (lopinavir, ritonavir, darunavir, atazanavir) on the SARS-CoV-2 Mpro/3CLpro and compared remdesivir with three orally administered NRTIs (tenofovir, FTC, 3TC, all of them as triphosphates) by docking on the SARS-CoV-2 RdRp. The authors found that all four protease inhibitors activated interactions with the key binding sites of Mpro/3CLpro and all three NRTIs were able to be incorporated as active forms in the same pocket where remdesivir or natural NTPs pose, with comparable inhibitory activity and activating similar interactions. The authors concluded by suggesting a combination of these drugs in future clinical trials for early treatment of outpatients with COVID-19. Finally, an interesting work by Salpini et al. [81] applied MDS to select the best inhibitors of SARS-CoV-2 Mpro/3CLpro and RdRp among FDA-approved drugs and then evaluated the impact of virus mutations on their binding affinity for the target enzymes. By virtual screening, the authors identified 20 potential inhibitors. Notably, while some selected drug candidates, like remdesivir, showed a decreased binding affinity for the P323L mutated RdRp, a prevalent variant observed together with the spike D614G variant in all continents except Asia, several other candidates including tenofovir were better recognized by the mutated enzyme. This study underlines the potentiality of SARS-CoV-2 variants to modulate drug susceptibility or resistance to these drugs.

Further studies identified tenofovir as a good candidate drug against SARS-CoV-2. In two studies, the spike/ACE2 interaction has been considered the target of simulated inhibition. MD and MDS analyses with 2080 FDA-approved molecules identified 238 compounds virtually binding the spike/ACE2 interaction interface, 10 of which were reported to have antiviral activity and with tenofovir and remdesivir showing the highest docking scores [82]. The authors found that remdesivir had stronger interactions with ACE2 than with spike, while TDF was closer to and interacted more strongly with the spike protein than the ACE2 receptor. Electrostatic energy mainly contributed to remdesivir interactions with ACE2, while both electrostatic and van der Waals interactions were responsible for TDF interactions with the spike protein. The authors also interestingly showed that the spike/ACE2 interaction was significantly decreased upon ligand binding, suggesting that those compounds may hinder the interactions between the spike and ACE2. Toor and colleagues [83] first identified the best spike/ACE2 virtual binding model based on a MM-GBSA approach and then docked 30 approved compounds against this model. Among these molecules, tenofovir showed significant molecular interactions with the ACE2 residues involved in the recognition of the spike protein. The potential inhibitory action of tenofovir was further validated by docking studies with the spike trimeric structure. By ADMET analysis, tenofovir also exhibited optimal availability and absence of predicted systemic toxicity.

Using MD of 61 antiviral drugs against five different co-crystallized forms of the Mpro/3CLpro of Sars-CoV-2, Sha and coworkers [84] found that, based on the docking score, tenofovir might be a good candidate compound with anti-SARS-CoV-2 effect. By MD and MDS analyses, Rahman and colleagues [85] virtually screened 29 FDA approved antiviral drugs against some promising target proteins and protein domains of SARS-CoV-2, such as the Mpro/3CLpro, the spike-receptor binding domain and the nsp9 RNA binding protein. The authors found that indinavir, sorivudine, cidofovir and darunavir had the highest binding affinity with viral proteins, and they also showed, when in complex with their target, resistance to deformation and low propensity for flexibility. By ADMET analysis, these drugs showed good bioavailability, distribution and metabolization but some toxic effects. Further, in order to find further potentially active drugs, the authors applied a homology virtual screening based on chemical similarity with the above drugs. They found that cidofovir predicted the structural analogue tenofovir as a similar approved drug, suggesting its use in further experimental studies.

Finally, Feng and colleagues [86] applied a novel in silico approach considering the energy contributions of each key residue involved in the binding between a target protein and its ligands. The authors determined the residue energy contributions in the binding pocket of SARS-CoV-2 Mpro/3CLpro toward binding the known inhibitor N3, and the total energy contribution of key residues around the ligand was used to construct an energy contribution vector of the protein. Using the generated recognition pattern of the protein, they performed a virtual screening with 1814 FDA-approved drugs and found 10 candidates, including TDF, with high pattern similarity to N3 and thus predicted to bind SARS-CoV-2 Mpro/3CLpro.

### 3.2. In Vitro Studies

Tenofovir activity against SARS-CoV-2 has also been analyzed in vitro in a few peer-reviewed studies. At concentrations under 100 µM, tenofovir did not inhibit viral replication in VeroE6 cells at a multiplicity of infection (MOI) = 0.02, when administered 1 h prior to infection and up to 48 h post-infection; although, as underlined by the authors of the paper, nucleotide analogues like tenofovir require activation into their triphosphate forms by host kinases, whose activity may differ among cell types, suggesting further studies in human airway epithelial cells [87]. In a similar cell model (Vero CCL-81 cells), further experiments gave contradictory results, likely due to the different experimental conditions used by the researchers (different MOIs and pre-treatment of cells with drugs). Both TAF and TDF demonstrated no anti-SARS-CoV-2 activity in those cells infected at a MOI = 0.1 and treated with the drugs for 48 h, with a half maximal effective concentration EC_50_ > 20 µM [88], while Clososki and colleagues [89] demonstrated anti-SARS-CoV-2 activity of TDF in the same cell model, but in different experimental conditions. They pre-exposed sub-confluent monolayers of cells to tenofovir or TDF 3 to 90 µM for 24 h, with untreated cells as control, and then to SARS-CoV-2 at a MOI = 1 for 2 h. Cells were then cultured in the presence or absence (control cells) of tenofovir or TDF for a further 48 h. The authors found that TDF was already able to reduce released viral genome by approximately 15-fold at the lowest concentration, without showing detectable cytotoxicity, while tenofovir did not inhibit viral replication at the same concentrations, possibly because TDF has faster intracellular uptake and accumulation of the active form tenofovir diphosphate. 

In further studies, TAF was inactive against SARS-CoV2 infection in an in vitro model of A549 human alveolar epithelial cells overexpressing ACE2 (MOI = 0.025), with half maximal effective concentration EC_50_ > 10 µM [90], although TDF was not evaluated in these cells. 

An increased number of circulating neutrophils has been described in COVID-19 patients with poor clinical outcome and released neutrophil-derived extracellular traps (NETs) may contribute to tissue injury. Interestingly, Veras and colleagues [91] found higher levels of NETs in the plasma, tracheal aspirate and lung tissue in COVID-19 patients compared with healthy controls and demonstrated that, in vitro, COVID-19 patients’ blood neutrophils released higher levels of NETs. The authors also demonstrated that neutrophils isolated from the blood of healthy controls may be infected by SARS-CoV-2 at a MOI = 0.5–1 and that infected cells released NETs in a MOI-dependent manner, while TDF treatment abrogated the release of NETs and reduced intracellular viral load.

Finally, using polymerase extension experiments, Chien and colleagues [92] demonstrated that tenofovir diphosphate was incorporated by SARS-CoV-2 RdRp into an RNA primer and terminated further polymerase extension, providing a molecular basis for inhibition of the SARS-CoV-2 RdRp by this nucleotide analog.

### 3.3. In Vivo Studies 

Only one in vivo study analyzed the efficacy of tenofovir, in combination with FTC, in an animal model [93]. Ferrets were inoculated with infective doses of the SARS-CoV-2 strain NMC-nCoV02 through the intranasal route and at one day post-infection (dpi) were administered daily with lopinavir/ritonavir, hydroxychloroquine sulfate or TDF/FTC for 14 days (10 ferrets for each group). Animals were also treated with phosphate-buffered saline (PBS) as control or with the immunosuppressive drug azathioprine for 7 days prior to infection. The intensity of clinical symptoms (fever, cough, rhinorrhea, reduced activity) of animals treated with anti-viral drugs was overall lower than in the PBS-treated group, while immunosuppressed animals showed similar but more persistent symptoms. Similar virus titers in nasal washes from treated animals were revealed at 2 to 8 dpi, with the exception of reduced titers at 8 dpi for TDF/FTC treated animals compared to the control group. In control and all antiviral-drug-treated animals, the virus was not detected at 10 dpi, while immunosuppressed animals showed delayed virus clearance. Viral RNA was also present in fecal specimens of all animals and in nasal turbinate tissue and lungs of three sacrificed animals per group. No differences in titer were observed over time for fecal specimens, while the virus persisted in nasal tissues after 8 dpi in all animals but with higher titers in immunosuppressed ferrets and only in the latter animals in the lungs. The anti-viral-treated animals showed neutralizing antibody titers similar to those of PBS-treated group until 14 dpi, but less at 21 dpi, while immunosuppressed animals had a persistently lower antibody response.

## 4. Clinical Data

### 4.1. Retrospective Studies

Boulle et al. [94] analyzed data in a big South-African cohort of HIV-infected patients (*n* = 536,574, of whom *n* = 3978 coinfected with SARS-CoV-2). Among those regularly receiving ART, authors found that TDF based-ART reduced COVID-19 mortality by 59% in HIV-SARS-CoV-2 coinfected patients vs. patients on ABC or ZDV. The association with reduced mortality remained after adjusting for kidney disease, viral suppression and ART duration (adjusted hazard ratio (aHR) 0.41: 95% CI, 0.31–1.04; *p* = 0.007). Also risk of hospitalization was reduced in patients receiving TDF/FTC vs. those taking ABC or ZDV, but without reaching statistical significance (aHR, 0.57: 95% CI, 0.21–0.78; *p* = 0.067). 

Moreover, in a further study on 77,590 HIV-infected people in Spain, 236 patients were diagnosed with COVID-19. In this cohort, TDF-based ART (TDF/FTC) also seemed to reduce the incidence of COVID-19 in HIV-infected persons compared to the general population (16.9% vs. 41.7%, respectively). No difference in COVID-19 incidence was confirmed when analysis was focused on patients using other ARVs (39.1% for TAF/FTC, 28.3% for ABC/3TC and 29.7% for others). No significant differences were found with regard to risk of hospitalization or death among HIV-infected patients on several ARVs (risk of hospitalization was 10.5% for TDF/FTC, 20.3% for TAF/FTC, 23.4% for ABC/3TC, and 20.0% for others) [95,96]. It is not easy to find the explanation for these results, but we must take into account several considerations: first, TAF is prescribed for patients who are less healthy that those prescribed with TDF; second, TDF reaches higher extracellular concentration with higher mucosal penetration; and third, possibly, it has a more potent immunomodulatory effect [97,98].

A recent meta-analysis [19] focused on the risk of several adverse COVID-19 outcomes in HIV-infected patients. The authors identified 11 studies that assessed the impact of ART on COVID-19 outcome in HIV-infected patients with no conclusive results (HIV-infected patients on ART with COVID-19, *n* = 4726, comprising the studies described above in [94,95,96]). First, studies included patients mostly on ART, therefore it was difficult to see the clear effect of ART, besides, patients receiving TDF were less clinically compromised than patients on others ARVs, which may explain the better COVID-19 outcome in HIV-infected patients receiving TDF.

Finally, evidence of the potential preventive effects of TDF or TAF on COVID-19 in HIV-negative people comes from PrEP users [99]. As stated in the Introduction section, TDF and TAF are efficaciously used as PrEP against HIV infection in people at risk of acquiring this infection [57,58,59,60,61]. In the study published by Ayerdi and colleagues [99], the prevalence of SARS-CoV-2 antibodies was higher in PrEP users (total *n* = 500) than that found in those who did not take the drug (total *n* = 250) (15.0% vs. 9.2%; *p* = 0.026), with lower prevalence of SARS-CoV-2 in users of TDF (total *n* = 409) than TAF (total *n* = 91), although without statistically significant differences (14.7% vs. 16.5%; *p* = 0.661).

### 4.2. Clinical Trials

The ClinicalTrials.gov webpage currently (12 April 2021) lists 8 trials (three planned, four currently recruiting and one completed) that investigate the efficacy of TDF or TAF in COVID-19 treatment or prophylaxis (https://clinicaltrials.gov accessed on 12 April 2021). Current clinical trials are listed in Table 1 [100,101,102,103,104,105,106,107].

## 5. Discussion and Conclusions

One year after the start of the pandemic, no oral antiretroviral drug is approved for preventing and/or treating SARS-CoV-2 infection. At the beginning of the pandemic, hydroxychloroquine [108] and lopinavir/ritonavir [8], two old drugs approved for other clinical uses, were extensively used for COVID-19 treatment but were quickly not recommended further [10,11,12,109]. However, remdesivir, which was tested early in the pandemic because of its wide range of antiviral activity against several RNA viruses and its safety profile, is currently the only antiviral medication approved for COVID-19 [7].

Today, it is not clear if other known drugs could potentially ameliorate the course of COVID-19 or even prevent SARS-CoV-2 infection. Finding a new therapeutic indication for an already known drug is a fast option because tolerability and safety have already been ascertained. As we have summarized in this review, clinical and pre-clinical data signal an intriguing and promising role of TDF in controlling the SARS-CoV-2 pandemic. As previously described with hydroxychloroquine and lopinavir/ritonavir, the extensive use of an old drug for COVID-19 as a new indication, may risk decreasing its offer to patients with the old indications. Moreover, their use can induce HIV resistance in patients with no known HIV-infection. TDF is a generic, inexpensive, well-known and widely available drug that is currently used for HIV and HBV treatment and for HIV prophylaxis [51,52,53,54,55,56,57,58,59,60,61,62,63,64,72]. It has good pharmacological characteristics with long serum and intracellular half-life, few adverse effects and drug–drug interactions and is orally administered [110]. No data on TDF efficacy in coronavirus infections other than COVID-19 are currently available, while information relating to SARS-CoV-2 infection is starting to come out. 

As reviewed in this work, several in silico studies suggest the inhibitory activity of tenofovir and its derivative prodrugs TDF and TAF on their presumed and more obvious viral target, the SARS-CoV-2 RdRp protein, likely competing with the natural nucleotide adenosine for the enzyme binding and then acting as chain terminators [50,73,74,75,76,77,78,79,80,81]. Notably, in one of these studies [81] a better binding activity of tenofovir with the P323L mutated RdRp protein was observed in comparison with remdesivir, further suggesting tenofovir use in COVID-19 treatment. Interestingly, these drugs also seem to bind other crucial viral proteins, such as PLpro and Mpro/3CLpro [84,85,86], and to interfere in the interaction between the S protein and ACE2 receptor [82,83], suggesting multiple targets in the virus life cycle for these drugs.

On the other hand, only a few in vitro studies on tenofovir/TDF/TAF efficacy have been described [87,88,89,90,91,92] and the results are contradictory, which is also due to the different experimental conditions. Therefore, further studies are needed to better evaluate the potential efficacy of this drug (alone or in combination with other antiviral drugs) against SARS-CoV-2 infection. Further, only one in vivo study analyzed the efficacy of tenofovir, in combination with FTC, in ferrets [93] and showed that TDF/FTC treatment reduces viral titers in nasal turbinate tissue, although the drug did not demonstrate efficacy in virus clearance in the gastrointestinal tract and lungs.

Clinical studies conducted thus far seem to suggest that TDF, but not TAF, based-ART may reduce COVID-19 incidence, risk of hospitalization and mortality in HIV-infected patients [19,94,95,96], while a preventive effect of TDF or TAF on COVID-19 is suggested by studies on HIV-negative PrEP users [99]. Taken together, although the available observational studies are few and do not provide conclusive data on TDF or TAF protective effects, the first compelling evidence suggests that more studies are needed to understand the role of this drug, both in COVID-19 treatment and prevention. Ongoing clinical trials [100,101,102,103,104,105,106,107] will add further valuable information on this topic.

In conclusion, although several studies have been conducted thus far on the action of antiretrovirals as therapy or prophylaxis against COVID-19 [11,111,112,113], the current clinical and pre-clinical evidence on the use of this drug deserves further investigation in the SARS-CoV-2 field in well-designed trials.

## Figures and Tables

**Figure 1 pharmaceuticals-14-00454-f001:**
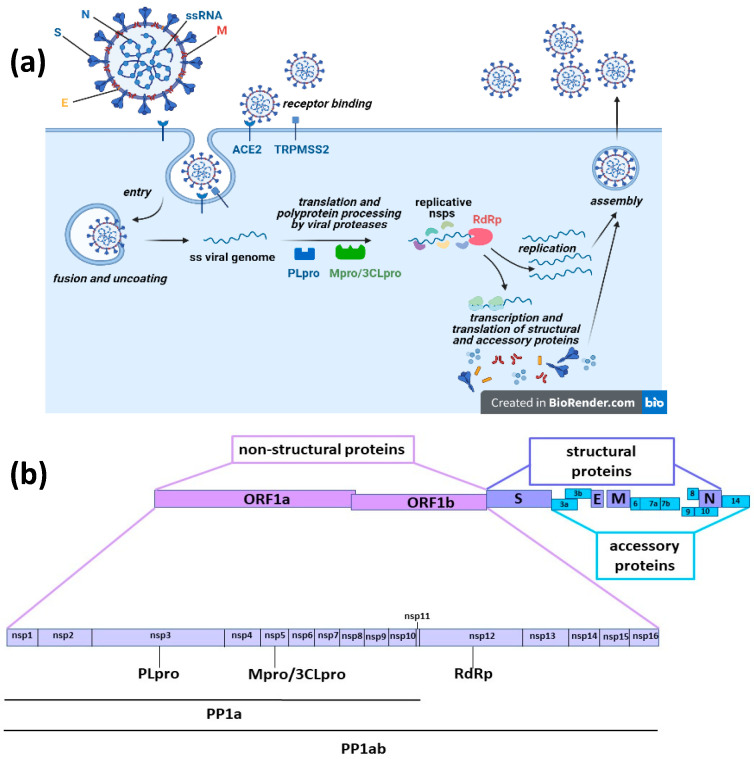
(**a**) SARS-CoV-2 is an enveloped positive-sense single-stranded RNA virus. In mature virions, structural proteins surround and interact with the viral genome (spike (S), envelope (E), membrane (M) and nucleocapsid (N) proteins). Virions enter target cells after specific binding of the receptor-binding domain (RBD) of the viral protein S to the cellular entry receptor, the angiotensin-converting enzyme 2 (ACE2). Viral uptake is promoted by the proteolytic activity of host factors, like the cell-surface transmembrane serine protease 2 (TMPRSS2), whose enzymatic activity is essential to permit the fusion with the cellular membrane and virus entry. Once entered target cells, virions are uncoated and viral RNA is released in the cytosol and immediately translated into two large polyproteins, pp1a and pp1ab, specified by two large open reading frames of the viral RNA, ORF1a and ORF1b. Both polyproteins contain the amino acid sequences of viral non-structural proteins (nsps). pp1a contains nsp1 to nsp11, while pp1ab contains nsp1 to nsp10 and nsp12 to nsp16. These polyproteins are co-translationally and post-translationally processed through proteolytic cleavage by two viral cysteine proteases residing in the nsp3 and nsp5 sequence, the papain-like protease (PLpro) and the main protease or 3C-like protease (Mpro/3CLpro), respectively. Released nsp1 inhibits host mRNA translation, while nsp2 to nsp16 are involved in the intracellular viral replication cycle; in particular, nsp12 is the RNA dependent RNA polymerase (RdRp) that with the cofactors nsp7 and nsp8 performs viral RNA synthesis for translation and synthesis of structural and accessory proteins and for the assembly of new virions. (**b**) SARS-CoV-2 genome and protein organization.

**Figure 2 pharmaceuticals-14-00454-f002:**
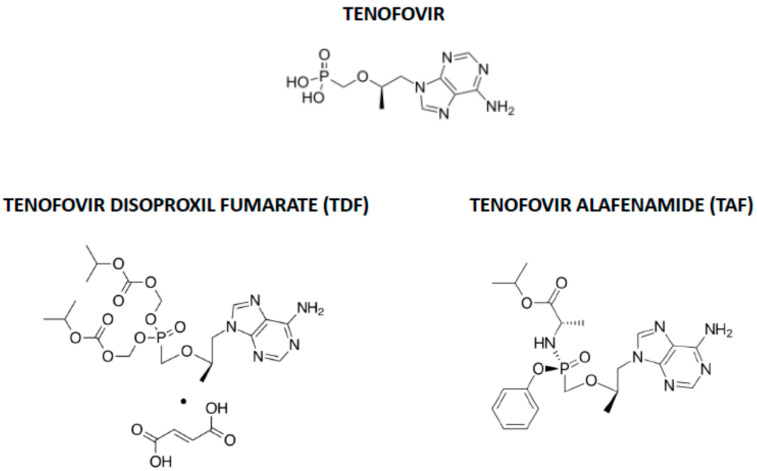
Chemical structures of tenofovir, tenofovir disoproxil fumarate (TDF) and tenofovir alafenamide (TAF).

**Figure 3 pharmaceuticals-14-00454-f003:**
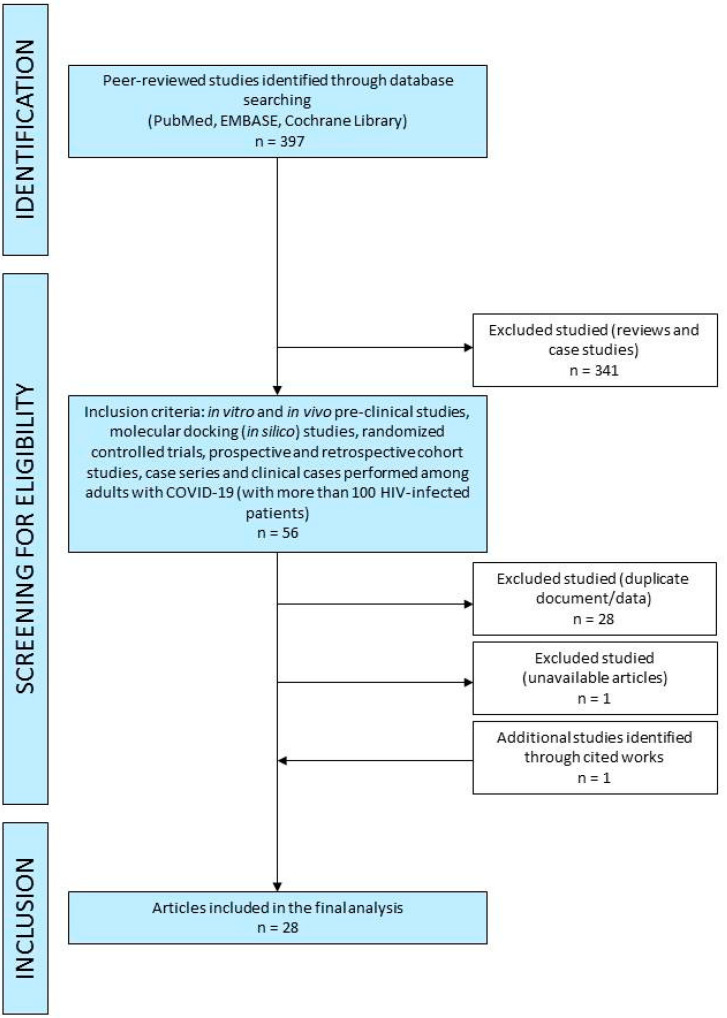
Flow chart of database searching and screening of studies for the systematic review.

**Table 1 pharmaceuticals-14-00454-t001:** Clinical trials currently investigating TDF/TAF efficacy in COVID-19 treatment or prophylaxis.

NCT Number [Ref]	Title	Status
NCT04519125 [100]	Daily Regimen of Tenofovir/Emtricitabine as Prevention for COVID-19 in Health Care Personnel in Colombia	not yet recruiting
NCT04405271 [101]	TAF/FTC for Pre-exposure Prophylaxis of COVID-19 in Healthcare Workers (CoviPrep Study)	not yet recruiting
NCT04575545 [102]	Prevalence of COVID-19 Infection in a Cohort of Patients Infected by the HIV and Patients Taking PrEP.	not yet recruiting
NCT04712357 [103]	Clinical Experimentation with Tenofovir Disoproxyl Fumarate and Emtricitabine for COVID-19 (ARTAN-C19)	recruiting
NCT04685512 [104]	Effect of Tenofovir/Emtricitabine in Patients Recently Infected With SARS-COV2 (Covid-19) Discharged Home (AR0-CORONA)	recruiting
NCT04359095 [105]	Effectiveness and Safety of Medical Treatment for SARS-CoV-2 (COVID-19) in Colombia	recruiting
NCT04334928 [106]	Randomized Clinical Trial for the Prevention of SARS-CoV-2 Infection (COVID-19) in Healthcare Personnel	recruiting
NCT04812496 [107]	Tenofovir-DF Versus Hydroxychloroquine in the Treatment of Hospitalized Patients With COVID-19 (TEDHICOV)	completed but no results posted

## Data Availability

Not applicable.

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
