# Peer review of "Tenofovir, Another Inexpensive, Well-Known and Widely Available Old Drug Repurposed for SARS-COV-2 Infection"

_pharmaceuticals, 2021, doi:10.3390/ph14050454_

Round 1
Reviewer 1 Report
The manuscript entitled “Tenofovir, another inexpensive, well known and widely available old drug repurposed for SARS-COV-2 infection” is a systematic review on the possible application of anti-HIV drug Tenofovir for the treatment of SARS-COV-2. It summarizes the most relevant in silico, in vitro, and in vivo results in this field, as well as some clinical data. This review can be a useful summary of the efforts made in the use of Tenofovir for the treatment of SARS-COV-2. The article is written in an appropriate way and the data are presented appropriately. However, I think that some changes should be made in order to improve the quality of the manuscript:
- The main drawback of the manuscript is the lack of more information and more references about the topic, especially in the introduction: it must be extended and more references should be included supporting some of the statements in the manuscript. For instance, the fourth paragraph (Lines 77-81) states that “It is not known if people living with HIV (PLWH) who are clinically and virologically stable could experience any greater risk for COVID-19 complications than the population without HIV infection and there is a possible protective role of antiretroviral therapy (ART)” without literature-support.
- The in silico section deals with many of the virus proteins as targets for drug interaction, which is really interesting. A paragraph must therefore be included in the introduction detailing the mechanism of infection of SARS-COV-2 with the appropriate references, indicating the role of those possible targets. I also encourage the authors to include a figure that illustrates it.
- As TFV is approved for HIV prevention and treatment, I encourage the authors to search and briefly indicate how the COVID pandemic affects people living with HIV.
- Since TFV, TAF and TDF are the main interest in this work, I think that the paragraph talking about them must be extended. For instance, TFV has previously been explored for its use in the treatment of other viral infections -such as HSV-, which could indicate that this is a multipurpose drug, as the manuscript is trying to demonstrate. Besides, they should indicate the current research status and applications of this drug and its prodrugs. For instance, it could be really interesting highlighting that they are frequently used for Preexposure prophylaxis (PrEP) of HIV, and if they are actually effective against SARS-COV-2, the treatment would offer a dual protection, really useful in countries where the access to the vaccine is hindered.
- The preclinical studies section should be rearranged. It would be way more logical starting with the in silico studies, as this is the first step to find the interaction of a drug with a target protein. Once the drug has proven able to interact with one (or more) targets, in vitro studies are useful to demonstrate that the drug will be able to interact with the viral target within the cell environment. After all of this, in vivo studies are useful to understand the effectivity and security of the drug after administered to a living being. I highly recommend this arrangement for the manuscript.
Some more specific comments are included below:
- Lines 39-41: Several vaccines have already been approved. However, it is arguable if they will reach all the population in the world, and more strategies are indeed needed. You should indicate the advantages that TFV and its prodrugs could bear. For example, could some patients get benefits from this if they cannot be vaccinated due to a health condition? It is interesting deepening this topic.
- Figure 2: Please, move this figure below “selection of studies” section, it is difficult to understand the figure if the section has not been read yet.
- Lines 167-178: These results seem contradictory. Please, explain and discuss it.
- Lines 235-237: Does TFV (or TDF, TAF) have any advantage compared with those three drugs? If not, why not choosing one of them instead?
- Line 273-320: According to these lines, TFV may interact not only with the RdRp, but also with other proteins. This is really interesting. Has the mechanism of interaction with those target proteins been described? If so, could the authors include it in the manuscript?
- Section 4.1.: Please, indicate the number of patients in the studies. This is necessary to understand the significance of the statistical differences; it is not the same 30 samples as 3000.
- Lines 354-358: As mentioned before, please, explain how HIV PrEP could also be effective against SARS-COV-2 and its potential benefits.
- Section 4.2.: Please, use a table to show the results instead of a list, and send the links to the “References” section.
- Lines 392-407: This is really interesting, but not actually a conclusion. Please, include it in the introduction section with appropriate literature supporting; the conclusion must be concise and highlight the most important ideas extracted from the results.
- The paragraphs in lines 155-157, 195-197, 321-323 and 359-362 are actual conclusions, and I recommend including them in this section.
Author Response
We wish to thank the Reviewer for the appreciation of our work and for thoughtful comments and constructive suggestions. We have now revised the manuscript accordingly, as described in detail below. We provide a clean version of the revised manuscript (“Revised manuscript”) and for the reviewer’ convenience a version with changes highlighted in red (“Revised manuscript with comments”).
Below, you can find Reviewers’ comments (in bold) and authors’ reply point by point.
1.The main drawback of the manuscript is the lack of more information and more references about the topic, especially in the introduction: it must be extended and more references should be included supporting some of the statements in the manuscript. For instance, the fourth paragraph (Lines 77-81) states that “It is not known if people living with HIV (PLWH) who are clinically and virologically stable could experience any greater risk for COVID-19 complications than the population without HIV infection and there is a possible protective role of antiretroviral therapy (ART)” without literature-support.
As kindly suggested, we have added more information and references supporting our statements in the Introduction, highlighted in red in the “Revised manuscript with comments” (see also reply to points 2, 3, 4, 6).
2.The in silico section deals with many of the virus proteins as targets for drug interaction, which is really interesting. A paragraph must therefore be included in the introduction detailing the mechanism of infection of SARS-COV-2 with the appropriate references, indicating the role of those possible targets. I also encourage the authors to include a figure that illustrates it.
As kindly asked by the reviewer (see also reply to point 1), we have added information on the mechanism of infection of Sars-CoV-2, illustrating the main viral proteins that could be possible targets of antiviral drugs, also adding a figure (lines 56-71 and Figure 1 revised version). In this paragraph we briefly describe virus life cycle and viral main proteins, referring the readers to some detailed reviews on the topic (references 4 to 6).
3.As TFV is approved for HIV prevention and treatment, I encourage the authors to search and briefly indicate how the COVID pandemic affects people living with HIV.
We thank the reviewer for the suggestion, however we think that this point is already covered enough in the introduction (see the three paragraphs beginning with “Generally, immunocompromised patients….”, “Updated to today, it is not clear…….” and “It is not known if people….”, lines 107-129 revised manuscript). Having added more information in the Introduction as kindly suggested (points 2, 4, 6), we fear that the introduction section becomes too long. We have however added more references, as suggested at point 1.
4.Since TFV, TAF and TDF are the main interest in this work, I think that the paragraph talking about them must be extended. For instance, TFV has previously been explored for its use in the treatment of other viral infections -such as HSV-, which could indicate that this is a multipurpose drug, as the manuscript is trying to demonstrate. Besides, they should indicate the current research status and applications of this drug and its prodrugs. For instance, it could be really interesting highlighting that they are frequently used for Preexposure prophylaxis (PrEP) of HIV, and if they are actually effective against SARS-COV-2, the treatment would offer a dual protection, really useful in countries where the access to the vaccine is hindered.
Thank you for the suggestion, we have extended the paragraph regarding these molecules within the Introduction, adding appropriate references (lines 130-168 revised manuscript).
5.The preclinical studies section should be rearranged. It would be way more logical starting with the in silico studies, as this is the first step to find the interaction of a drug with a target protein. Once the drug has proven able to interact with one (or more) targets, in vitro studies are useful to demonstrate that the drug will be able to interact with the viral target within the cell environment. After all of this, in vivo studies are useful to understand the effectivity and security of the drug after administered to a living being. I highly recommend this arrangement for the manuscript.
We agree and apologize, we have now rearranged this section as suggested.
6.Lines 39-41: Several vaccines have already been approved. However, it is arguable if they will reach all the population in the world, and more strategies are indeed needed. You should indicate the advantages that TFV and its prodrugs could bear. For example, could some patients get benefits from this if they cannot be vaccinated due to a health condition? It is interesting deepening this topic.
Thanks to the reviewer for raising this important point. We have added some considerations regarding the advantages of efficacious therapies against Sars-CoV-2 infection, extending this consideration not only to TFV but, broadly speaking, to antiviral drugs (lines 43-51, revised manuscript).
7.Figure 2: Please, move this figure below “selection of studies” section, it is difficult to understand the figure if the section has not been read yet.
Thanks, we have moved it.
8.Lines 167-178: These results seem contradictory. Please, explain and discuss it.
We apologize for the confuse explanation in this paragraph. We have now better explained the two studies and discussed the apparently contradictory results, likely due to different experimental conditions (different MOI, pre-treatment of cells, etc) (lines 344-358, revised manuscript).
9.Lines 235-237: Does TFV (or TDF, TAF) have any advantage compared with those three drugs? If not, why not choosing one of them instead?
Authors of the study decided to conduct further MDS analyses only on the 3 best-ranking drugs penciclovir, ribavirin and zanamivir, based on binding strength, stability of the virtual interaction and steric hindrance. In this way unfortunately they excluded tenofovir for further analysis (at least published analysis), although it demonstrated good binding scores.
10.Line 273-320: According to these lines, TFV may interact not only with the RdRp, but also with other proteins. This is really interesting. Has the mechanism of interaction with those target proteins been described? If so, could the authors include it in the manuscript?
We agree with the reviewer in considering surprising, worth and interesting the interactions of tenofovir and its prodrugs with target proteins that are so different from the natural target (i.e. RdRp), like Mpro and PLpro. However, in all docking studies, for receptor/ligand-binding energy calculation researchers take into account several types of chemical interactions between a protein and a drug/molecule (or more broadly, a receptor and a ligand), i.e. electrostatic interactions, like hydrogen bonds, and van der Waals interactions and they frequently consider not only the natural targets. Then those are the mechanism of interaction (different inter-molecular chemical bonds) between the target protein and studied ligand, that are described in the cited papers and, when described, also reported by us in this review.
11.Section 4.1.: Please, indicate the number of patients in the studies. This is necessary to understand the significance of the statistical differences; it is not the same 30 samples as 3000.
Correct, we have added the number of patients in the text, as kindly suggested.
12.Lines 354-358: As mentioned before, please, explain how HIV PrEP could also be effective against SARS-COV-2 and its potential benefits.
Thanks again for the suggestion. We have clarified in this paragraph how HIV PrEP could also be effective against COVID-19 (lines 436-443 revised manuscript), referring the readers to the modified paragraph in the Introduction section (lines 130-168 revised manuscript).
13.Section 4.2.: Please, use a table to show the results instead of a list, and send the links to the “References” section.
Thanks for the suggestion, we have added the table (Table 1), sending the links to the References sections.
14.Lines 392-407: This is really interesting, but not actually a conclusion. Please, include it in the introduction section with appropriate literature supporting; the conclusion must be concise and highlight the most important ideas extracted from the results.
The paragraphs in lines 155-157, 195-197, 321-323 and 359-362 are actual conclusions, and I recommend including them in this section.
We agree and thank the reviewer for the correct suggestions. We have modified the “Conclusions” section with the name “Discussion and Conclusions”, not to stretch too much the “Introduction” section after the addition of requested points (see points 1, 2, 4 and 6), also briefly discussing the current evidence observed in in silico, in vitro, in vivo and clinical studies.
Reviewer 2 Report
In this paper, Zanella and coworkers reviewed the currently available evidence on tenofovir efficacy against SARS-CoV-2 infection. They performed literature survey including in vivo, in vitro, in silico, and clinical studies on tenofovir against SARS-COV-2 infection.
This is well reviewed, however, I think there are some issues to be clarified.
- It is easy to understand if the clinical trial part is summarized as a table with more information.
- Although tenofovir is a potent antiviral agents, recently, the TDF-resistant quadruple mutations have been identified (http://dx.doi.org/10.1016/j.jhep.2019.02.006) in HBV reverse transcriptase. Please include the tenofovir-resistant virus/mutations to provide up-to-date on tenofovir treatment.
- There is typo in Abstract. “higher mortality in people leaving with HIV (PLWH)”
- According to Figure 2, reviews and case studies are excluded in this study. I think the reviews and case studies also can provide lots of information on this subject. Therefore, I think these studies should be included in this manuscript.
- For example, include a recent comprehensive literature review which investigated similar study with this manuscript (The Journal of Clinical Pharmacology 2020, DOI: 10.1002/jcph.1788)
Author Response
We wish to thank the Reviewer for the appreciation of our work and for thoughtful comments and constructive suggestions. We have now revised the manuscript accordingly, as described in detail below. We provide a clean version of the revised manuscript (“Revised manuscript”) and for the reviewers’ convenience a version with changes highlighted in red (“Revised manuscript with comments”).
Below, you can find Reviewers’ comments (in bold) and authors’ reply point by point.
1.It is easy to understand if the clinical trial part is summarized as a table with more information.
Thanks for the suggestion, we have added the table (Table 1), sending the links to the References sections.
2.Although tenofovir is a potent antiviral agents, recently, the TDF-resistant quadruple mutations have been identified (http://dx.doi.org/10.1016/j.jhep.2019.02.006) in HBV reverse transcriptase. Please include the tenofovir-resistant virus/mutations to provide up-to-date on tenofovir treatment.
Thank you for the suggestion, we have added a paragraph explaining more in depth the molecules and their clinical use, focusing also on drug resistance.
3.There is typo in Abstract. “higher mortality in people leaving with HIV (PLWH)”
We understand that this sentence may be misinterpreted, we have changed it with “higher mortality in HIV-infected people”. We thank the reviewer for noticing.
4.According to Figure 2, reviews and case studies are excluded in this study. I think the reviews and case studies also can provide lots of information on this subject. Therefore, I think these studies should be included in this manuscript. For example, include a recent comprehensive literature review which investigated similar study with this manuscript (The Journal of Clinical Pharmacology 2020, DOI: 10.1002/jcph.1788)
We thank the reviewer for the suggestions. We excluded case reports since we gave priority to studies with at least 100 patients/subjects, for a more statistical approach. We recognize that also case reports are important for clinicians and pharmacologists, however they are difficult to discuss with the statistical approach we used in this part of the review. We have however cited a meta-analysis [ref. 19] comprising reports with also a number of patients < 100. Regarding reviews, we have considered them during the collection of articles to retrieve those articles that could be missed during searching of PubMed, EMBASE and Cochrane Library. In this way we indeed retrieved the article described in Figure 3 (additional studies identified through cited works n =1). We have read with great interest also the review you suggested (The Journal of Clinical Pharmacology 2020, DOI: 10.1002/jcph.1788) and we have cited it in the “Discussion and Conclusions” section [ref 114].
Round 2
Reviewer 1 Report
The authors have responded adequately to the suggestions and questions I made. The article is much better organized now and the information is more complete. Thank you so much for taking my humble contributions into consideration.